# Interplay of Impaired Cellular Bioenergetics and Autophagy in PMM2-CDG

**DOI:** 10.3390/genes14081585

**Published:** 2023-08-04

**Authors:** Anna N. Ligezka, Rohit Budhraja, Yurika Nishiyama, Fabienne C. Fiesel, Graeme Preston, Andrew Edmondson, Wasantha Ranatunga, Johan L. K. Van Hove, Jens O. Watzlawik, Wolfdieter Springer, Akhilesh Pandey, Eva Morava, Tamas Kozicz

**Affiliations:** 1Department of Clinical Genomics, Mayo Clinic, Rochester, MN 55905, USA; 2Department of Laboratory Medicine and Pathology, Systems Biology and Translational Medicine Laboratory, Mayo Clinic, Rochester, MN 55905, USA; 3Department of Neuroscience, Mayo Clinic, Jacksonville, FL 32224, USA; 4Neuroscience PhD Program, Mayo Graduate School of Biomedical Sciences, Mayo Clinic, Jacksonville, FL 32224, USA; 5Department of Pediatrics, Division of Human Genetics, Children’s Hospital of Philadelphia, Philadelphia, PA 19104, USA; 6Department of Pediatrics, Section of Clinical Genetics and Metabolism, University of Colorado, Aurora, CO 80309, USA; 7Manipal Academy of Higher Education, Manipal 576104, Karnataka, India; 8Department of Biophysics, University of Pecs Medical School, 7624 Pecs, Hungary; 9Department of Anatomy, University of Pecs Medical School, 7624 Pecs, Hungary

**Keywords:** LC3-II autophagy marker, congenital disorders of glycosylation, phosphomannomutase 2 deficiency, secondary suboptimal mitochondrial function, mitophagy, glycoproteomics, proteomics

## Abstract

Congenital disorders of glycosylation (CDG) and mitochondrial disorders are multisystem disorders with overlapping symptomatology. Pathogenic variants in the PMM2 gene lead to abnormal N-linked glycosylation. This disruption in glycosylation can induce endoplasmic reticulum stress, contributing to the disease pathology. Although impaired mitochondrial dysfunction has been reported in some CDG, cellular bioenergetics has never been evaluated in detail in PMM2-CDG. This prompted us to evaluate mitochondrial function and autophagy/mitophagy in vitro in *PMM2* patient-derived fibroblast lines of differing genotypes from our natural history study. We found secondary mitochondrial dysfunction in PMM2-CDG. This dysfunction was evidenced by decreased mitochondrial maximal and ATP-linked respiration, as well as decreased complex I function of the mitochondrial electron transport chain. Our study also revealed altered autophagy in PMM2-CDG patient-derived fibroblast lines. This was marked by an increased abundance of the autophagosome marker LC3-II. Additionally, changes in the abundance and glycosylation of proteins in the autophagy and mitophagy pathways further indicated dysregulation of these cellular processes. Interestingly, serum sorbitol levels (a biomarker of disease severity) and the CDG severity score showed an inverse correlation with the abundance of the autophagosome marker LC3-II. This suggests that autophagy may act as a modulator of biochemical and clinical markers of disease severity in PMM2-CDG. Overall, our research sheds light on the complex interplay between glycosylation, mitochondrial function, and autophagy/mitophagy in PMM2-CDG. Manipulating mitochondrial dysfunction and alterations in autophagy/mitophagy pathways could offer therapeutic benefits when combined with existing treatments for PMM2-CDG.

## 1. Introduction

Congenital Disorders of glycosylation (CDG) is a group of more than 160 genetic disorders that affect protein and lipid glycosylation [1]. CDG and mitochondrial disorders are multisystem disorders with clinical characteristics that overlap [2,3]. Aberrant mitochondrial physiology has been reported in patients with ST3GAL5-CDG (blood lactate elevation, suboptimal respiratory chain complex III function in fibroblasts and muscle [4]), PMM2-CDG [5] (lactate elevation in brain during stroke-like episodes), PGM1-CDG [6] (elevated lactic acid in blood, and altered complex III activity in fibroblasts [7]), SLC39A8-CDG [8] (Leigh-like brain MRI, complex deficiencies and elevated lactate), and PIGP-CDG (reduced complex I activity in muscle, reduced pyruvate dehydrogenase complex activity in muscle and fibroblasts [9]). However, whether impaired cellular bioenergetics characterizes other CDG is still elusive. This has prompted us to test the hypothesis that PMM2-CDG would present with secondary suboptimal mitochondrial function. This is important as secondary suboptimal mitochondrial function in PMM2-CDG could modify disease onset and severity of symptoms.

PMM2-CDG patients present with a wide range of clinical features, including hypotonia, global developmental delay, seizures, stroke-like episodes, liver disease, and endocrine and hematologic abnormalities [10]. Phosphomannomutase-2-CDG (PMM2-CDG) is the largest CDG described up to date (>900 patients reported) [10]. PMM2-CDG has an incidence between 1:20,000 and 1:100,000 [10]. It is caused by pathogenic variants in *PMM2*, resulting in a deficiency of PMM2 enzyme. This essential enzyme is involved in the conversion of mannose-6-P (man-6-P) into mannose-1-P (man-1-P), and the block in PMM2 triggers a depletion of Man-1-P and its downstream metabolites GDP-mannose and Dolichol-P-mannose (Dol-Man) [10].

Pathogenic variants in the *PMM2* gene lead to abnormal N-linked glycosylation [10]. Abnormal protein glycosylation in CDG results in the buildup of misfolded proteins that results in endoplasmic reticulum (ER) stress [11,12,13,14,15]. This has been corroborated in patient-derived fibroblasts and models of PMM2-CDG, which presented with increased unfolded protein response (UPR) and chronic ER stress [16,17,18]. ER stress has also been shown to induce autophagy, and in PMM2-CDG, an increase in autophagy has been reported in genetically engineered PMM2 model cells [19]. UPR response and ER stress have also been connected to secondary suboptimal mitochondrial function [20,21].

The primary function of autophagy does not seem to be mediating programmed cell death, but autophagy has a clear pro-survival role during ER stress as it acts as an alternative mechanism for the clearance of misfolded or damaged proteins that cannot be cleared by the UPR [22]. In line with this notion, autophagy functions across a diverse range of species as a pro-survival pathway during various forms of cellular stress [19,22,23,24]. Although protein degradation is a salient feature of autophagy, studies over the past decade have also revealed that autophagy plays a key part in mobilizing diverse cellular energy [22,25]. In addition, autophagy is now recognized as a critical housekeeping pathway and a form of metabolic adaptation. Abnormalities in autophagy have been associated with various diseases, such as neurodegenerative and metabolic disorders [19,26,27,28]. The growing appreciation of the significance of autophagy in disease prompted us to test the hypothesis that autophagy would be altered in PMM2-CDG and that autophagy could act as a modulator of biochemical and clinical markers of disease severity.

In this study, we conducted a comprehensive analysis of fibroblasts derived from PMM2-CDG patients to investigate the potential association of PMM2-CDG with secondary mitochondrial dysfunction and abnormalities in autophagy/mitophagy pathways. Our findings revealed evidence of secondary mitochondrial dysfunction and identified changes in the abundance of several proteins and glycoproteins associated with autophagy/mitophagy pathways. These results indicate that the variability in genotype–phenotype associations and the significant clinical variations in multi-organ involvement seen in PMM2-CDG could, at least in part, be influenced by impaired mitochondrial function and autophagy/mitophagy.

## 2. Patients and Methods

### 2.1. Prospective Clinical and Biochemical Data

We evaluated the genetic, laboratory, metabolic, and clinical data of 8 patients with PMM2-CDG (Table 1) enrolled in the Frontier in CDG Consortium (FCDGC) natural history study (IRB: 19–005187; NCT04199000). All skin biopsies were obtained for establishing fibroblasts as part of the standard clinical care. Disease severity was assessed using the Nijmegen Pediatric CDG Rating Scale (NPCRS), most severe = 82; mild = 0–14, moderate = 15–25, and severe = >26 [29,30]. P3, P7, and P8 were previously reported by Ligezka et al. [31].

#### Cell Culture

Skin fibroblasts from nine patients with PMM2-CDG and five age-matched control fibroblasts (GM5400 GM5381 GM8398 GM5757 and GM0038; Coriell Institute, Camden, NJ, USA) were maintained at subconfluent densities in Minimum Essential Media (MEM; Gibco, Billings; MT, USA, 1 g/L glucose) supplemented with 10% Fetal Bovine Serum (FBS; Gibco), 10% antibiotic–antimycotic with gentamicin, and 10% non-essential amino acids at 37 °C (with 5% CO_2_). The medium was changed 24 h before harvest. For autophagy flux detection, chloroquine (Sigma-Aldrich, St. Louis, MO, USA; Cat# C6628) and rapamycin (Med Chem Express, Monmouth Junction, NJ, USA; Cat# HY-10219) were added into the medium 24 h before harvest. Cells were harvested by scraping in phosphate-buffered saline (PBS, pH7.4) and centrifuged at 2000 rpm at 4 °C for 10 min. 

### 2.2. PMM Activity, and Serum Transferrin and Polyol Levels

The methods for measuring PMM activity, polyol levels and transferrin have been discussed in Ligezka et al. 2022 [31]. PMM and phosphomannose isomerase (MPI) activity was assayed via spectrophotometric measurements. Sorbitol and mannitol levels were determined using gas chromatography/mass spectrometry (GC/MS). The degree of glycosylation abnormality was described via transferrin glycoform analysis (Affinity Chromatography-Mass Spectrometry, MS).

### 2.3. Western Blot Analysis

Protein samples for Western blot analysis were prepared as previously described. [19]. Briefly, harvested cells were lysed with RIPA lysis buffer (50 mM Tris–HCl, pH 8.0, with 150 mM sodium chloride, 1.0% NP–40, 0.25% sodium deoxycholate, and 0.1% sodium dodecyl sulfate, Sigma-Aldrich) supplemented with protease inhibitors (Sigma- Aldrich) and centrifuged at 1400 rpm at 4 °C for 30 min. Protein concentrations were estimated by using BCA assay (Thermo Fischer Scientific, Waltham, MA, USA). Equal amounts of proteins were resolved via SDS-PAGE on a Novex 10% Bis-Tris Gel (Thermo Fischer Scientific) and transferred onto a nitrocellulose membrane using transfer stacks in the iBlot2 Gel Transfer Device (Invitrogen, Waltham, MA, USA). The membranes were blocked for 1 h in Fish Serum Blocking Buffer (Thermo Fischer Scientific, Waltham, MA, USA) at 4 °C and then incubated with rabbit anti-LC3B antibody (Novus Biologicals, Centennial, CO, USA; Cat# NBP2-46892) and mouse anti-β actin antibody (ABclonal, Woburn, MA, USA; Cat# AC004) as an internal control diluted in blocking buffer for 72 h at 4 °C. After brief washing, the membranes were incubated with Biotin-SP conjugated Anti-Rabbit IgG (Jackson Immuno Research, West Grove, PA, USA; Code# 711-065-152) for 1 h at 4 °C, and then incubated with Alexa Fluor^R^ 680 Streptavidin (Jackson Immuno Research; Code# 016-620-084) for 30 min at 4 °C. The membranes were visualized on a Licor Odyssey CLx Infrared Imaging System (LI-COR Biosciences, Lincoln, NE, USA.). Signal intensity was quantified offline by Odyssey software (version 2.0). All measured intensities of protein bands were normalized with b actin intensity.

### 2.4. Immunocytochemistry

In total, 8 × 10^4^ skin fibroblasts from each patient with PMM2 and age-matched control fibroblasts were seeded on chambers and fixed with 4% paraformaldehyde. Cells were blocked for 1 h in Blocker^TM^ Casein in PBS (Thermo Fischer). Then, cells were stained with rabbit anti-LC3 antibody (Novus Biologicals; Cat# NBP2-46892) overnight at 4 °C and visualized via incubation with anti-Rabbit IgG Alexa Flour^TM^ 488 (Thermo Fischer Scientific; Cat# A-21206) for 1 h at 4 °C. These fibroblasts were counterstained with DAPI (Vector Laboratories, Newark, CA, USA; Cat# H-1200). Images were acquired using Carl Zeiss laser scanning confocal microscope LSM780 and processed with Zen black software.

### 2.5. Mitochondrial Respiration

The Agilent Seahorse XF Cell Mito Stress Test was used to investigate the Oxygen Rate Consumption (OCR) of fibroblast cell lines. The OCR was assessed with the Seahorse XF96 Extracellular Flux Analyzer (Agilent, Santa Clara, CA, USA) using 10,000 cells per well seeded for 48 h, as previously described [32]. Before measuring respiration, the culture medium was replaced with XF Base Medium Minimal DMEM (102353-100, Agilent) supplemented with 10 mM glucose, 1 mM pyruvate, and 2 mM L-glutamine to measure mitochondrial respiration. After measuring the basal respiration, 2.5 μM oligomycin, 2.0 μM carbonyl cyanide phenylhydrazone (FCCP), and 0.5 μM rotenone + antimycin A were added sequentially. FCCP concentrations were titrated to determine an optimal concentration for a given experiment (2.0 μM FCCP). Each sample measurement was repeated 5 times in the 8 well replicates setting. Normalization was performed via a direct cell count after adding a cell membrane-permeable nuclear staining compound (Hoechst 33342) following automated imaging and cell counting. In addition, the citrate synthase activity was measured and applied for normalization to quantify the activity where a different number of mitochondria is present in one sample compared to the other samples. Citrate synthase is a matrix enzyme of the TCA/Krebs cycle that catalyzes the conversion of oxaloacetate and acetyl-CoA to citrate and CoA. Acetyl-CoA and oxaloacetate, DTNB (5’,5’dithiobis-(2nitrobenzoaat)) were added to form a complex with CoA that has an absorption spectrum at 412 nm. An increased absorption at this wavelength is a measure of CS activity. For a blank measurement, the absorption at 412 nm was measured without the substrate oxaloacetate. Seahorse XF real-time ATP rate assay was used to investigate the change in cellular metabolism. Cells were plated as described above in assay media and stimulated following the injection of 2.5 μM oligomycin and 0.5 μM rotenone + antimycin A. Seahorse XF real-time ATP rate assay was used to investigate the change in cellular metabolism. Cells were plated as described above in assay media and stimulated following the injection of 2.5 μM oligomycin and 0.5 μM rotenone + antimycin A.

### 2.6. Mitochondrial Electron Transport Chain (mtETC) Complex Enzymology

The activities of mitochondrial complexes I (CI), II (CII), III (CIII), and IV (CIV) were assayed using a spectrophotometric enzyme activity assay [33] performed on a FLUOstar Omega spectrophotometric plate reader (BMG). mtETC complex activities were measured and normalized to both citrate synthase (CS) activity and protein concentration. Cultured patient skin fibroblasts were trypsinized, collected, counted, and washed in phosphate-buffered saline (PBS). Cells were homogenized in 20 mM Tris-HCl (pH 7.6) (400 μL/5× 10^6^ cells) using a bead mill homogenizer and 1.5 mL microtubes pre-filled with 1.4 mm ceramic beads (Omni International, Kennesaw, GA, USA). 

#### 2.6.1. Complex I (CI)

Cell homogenates were incubated in a potassium phosphate (K_2_HPO_4_)-buffered solution of bovine serum albumin (BSA) in the presence of nicotinamide adenine dinucleotide (NADH), ubiquinone (CoQ), and the artificial electron receptor 2,6-dichlorophenolindophenol (DCPIP). Ubiquinone reduced to ubiquinol (QH_2_) via the oxidation of NADH to NAD^+^ by CI rapidly reduces DCPIP (blue) to DCIPIH_2_ (colorless). The NADH oxidation was assayed through spectrophotometric measurement of the extinction of DCPIP absorption at 600 nm over 17 min. Non-specific NADH oxidation was determined by simultaneously assaying NADH oxidation in each tissue homogenate in the presence of the potent CI inhibitor rotenone, and CI activity was calculated by subtracting the non-specific (rotenone-inhibited) NADH oxidation from the total (rotenone-uninhibited) NADH oxidation.

#### 2.6.2. Complex II (CII)

Cell homogenates were incubated in a K_2_HPO_4_-buffered solution of BSA, ethylenediaminetetraacetic (EDTA), and sodium azide (NaAz) in the presence of succinate, decylubiquinone (DUB), adenosine triphosphate (ATP) and DCPIP. DUB reduced to DUH_2_ via the oxidation of succinate by CII rapidly oxidizes DCPIP (blue) to DCIPIH_2_ (colorless). Succinate oxidation was assayed through spectrophotometric measurement of the extinction of DCPIP absorbance at 600 nm for 15 min. Non-specific reduction of DCPIP was corrected for by simultaneously assaying each tissue homogenate in the presence of the potent CII inhibitor malonate, and CII activity was calculated by subtracting the non-specific (malonate-inhibited) succinate oxidation from the total (malonate-uninhibited) succinate oxidation.

#### 2.6.3. Complex III (CIII)

Cell homogenates were incubated in a K_2_HPO_4_-buffered solution of EDTA, NaAz, and polysorbate 20 in the presence of reduced DUH_2_ and cytochrome C (CytC). Reduction of CytC by CIII via oxidation of DUH_2_ will be assayed via spectrophotometric measurement of reduced CytC absorbance at 550 nm over 15 min. Non-specific reduction of CytC will be corrected for by first assaying for CytC reduction in the presence of DUH2 but the absence of the homogenate.

#### 2.6.4. Complex IV (CIV)

Cell homogenates were incubated in a K_2_HPO_4_-buffered solution in the presence of reduced CytC. The oxidation of reduced CytC by CIV will be assayed via spectrophotometric measurement of the extinction of absorption by reduced CytC at 550 nm over 15 min. The reaction endpoint will be assessed by artificially oxidizing all reduced CytC in the reaction mixture via the addition of the potent oxidizer potassium ferricyanide (K_3_Fe(CN)_6_), and the total reduced CtyC in the reaction will be quantified using a triplicate of blank wells lacking tissue homogenate. 

#### 2.6.5. Citrate Synthase (CS)

Cell homogenates will be incubated in a Tris-HCl-buffered solution in the presence of oxaloacetic acid, acetyl-CoA, and 5,5′-dithiobis-(2-nitrobenzoic acid) (DTNB). CoA-SH generated via the condensation of acetyl-CoA and oxaloacetic acid by CS to citric acid rapidly cleaves DTNB (colorless) to TNB^−^, which rapidly ionizes to TNB^2−^ (yellow). Condensation of acetyl-CoA and oxaloacetic acid by CS will be assayed via spectrophotometric measurement of TNB^2−^ at 411 nm over 15 min.

#### 2.6.6. Protein Concentration

The protein concentrations of cell homogenates were assayed using the Pierce BCA Protein Assay (Millipore).

### 2.7. Phospho-Ubiquitin p-S65-Ub ELISA

Relative levels of the PINK1/PRKN mitophagy marker phosphorylated Ubiquitin (p-S65-Ub) were determined using ELISA, as described previously [34]. Briefly, 96-well ELISA plates (Mesoscale Diagnostics, Rockville, MD, USA, L15XA-6) were coated overnight with 1 µg/mL pS65-Ub antibody (Cell Signaling Technology, Danvers, MA, USA, 62802S) in 200 mM sodium carbonate buffer, pH 9.7. The plates were then blocked with 5% BSA in TBST for 1 h, and cell lysates were diluted in blocking buffer with a final concentration of 10 µg/well for 2 h. Plates were washed in TBST, and 5 µg/mL detecting Ub antibody (Thermo Fisher Scientific, 14-6078-82) was incubated for 2 h. After three washes, a sulfo-tag labeled secondary anti-mouse antibody (Mesoscale Diagnostics, R32AC-1) was added for detection. The assay was read in Gold Read buffer (Mesoscale Diagnostics, R92TG-2) on a MESO QuickPlex SQ 120 (Mesoscale Discovery, Rockville, MD, USA). The buffer blank was subtracted from the measured raw values.

### 2.8. Proteomic and Glycoproteomic Analysis

#### 2.8.1. Sample Preparation

Samples for proteomic and glycoproteomic analysis were prepared as previously described [35]. Briefly, fibroblasts cells from 3 PMM2-CDG patients (P3, P7 and P8) and 3 controls (GM5400, GM5381 and GM5757) were lysed in sonication. An equal amount of total protein from each serum sample was reconstituted in 8 M Urea in 50 mM triethylammonium bicarbonate (TEAB), pH 8.5. Proteins were reduced via incubation with dithiothreitol (Sigma) at a final concentration of 10 mM at 37 °C for 30 min and alkylated via incubation with 40 mM iodoacetamide (Sigma) for 30 min in the dark at RT. The samples were digested with trypsin (Worthington, TPCK treated) at a ratio of 1:20 (trypsin:total protein) at 37 °C overnight.

The resulting peptides were labeled with tandem mass tag (TMTPro, Thermo Fisher) reagents as per the manufacturer’s instructions and subsequently pooled.

About 80% of dried TMT-labeled peptides were used for the glycopeptide enrichment via size-exclusion chromatography (SEC), as described previously [36]. Isocratic flow with 0.1% formic acid was used for SEC, and 12 fractions were collected over a run time of 130 min. The remaining 10% of dried TMT-labeled peptides were fractionated via basic pH reversed-phase liquid chromatography (bRPLC) for proteomics on a C18 column using an Ultimate 3000 UHPLC System. Then, 5 mM ammonium formate in water, pH 9,and 5 mm ammonium formate in 90% acetonitrile, pH 9, were used as solvent A and B, respectively. Ninety-six fractions collected over 120 min were concatenated into 12 fractions.

#### 2.8.2. Liquid Chromatography–Tandem Mass Spectrometry (LC-MS/MS) Analysis

LC-MS/MS analysis of fractionated samples from both proteomics and glycoproteomics was carried out as previously described [35] with some modifications. Briefly, 12 fractions from SEC and 12 concatenated fractions from bRPLC were analyzed by Orbitrap Eclipse mass spectrometer (Thermo Fisher Scientific). Peptides were separated via liquid chromatography on an EASY-Spray column (75 m × 50 cm, PepMap RSCL C18, Thermo Fisher Scientific). Then, 0.1% formic acid in water (solvent A) and 0.1% formic acid in acetonitrile (solvent B) were used as solvents. Peptides were trapped on a trap column (100 mm × 2 cm, Acclaim PepMap100 Nano-Trap, Thermo Fisher Scientific) at a flow rate of 20 µL/min. LC separation was performed at a flow rate of 300 nL/min using a 150 LC gradient. All experiments were performed in the DDA mode, with the top 15 ions isolated at a window of 0.7 *m*/*z* and a default charge state of +2. Precursor ions were acquired at a resolution of 120,000 (at *m*/*z* 200) for precursor ions and at a resolution of 30,000 (at *m*/*z* 200) for fragment ions. The fragmentation was carried out using the higher-energy collisional dissociation (HCD) method using a normalized collision energy of 34 for proteomics or stepped HCD (15, 25, 40) for glycoproteomics.

#### 2.8.3. Data Analysis

Data analysis was performed as described previously [35]. The proteomics data were searched using the Sequest search engine in Proteome Discoverer 2.5 against the Uniprot Human Reviewed protein sequences and the glycoproteomics data using the publicly available software pGlyco version 3 with an in-built glycan database [37]. Two missed cleavages were allowed for both proteomics and glycoproteomics analysis. Error tolerance for precursor and fragment ions were set to 10 ppm and 0.02 Da, respectively, for proteomics and 10 ppm and 20 ppm for fragment ions, respectively, for glycoproteomics. Cysteine carbamidomethylation was set as a fixed modification, whereas the oxidation of methionine and the deamidation of asparagine were set as variable modifications. The false discovery rate (FDR) was set to 1% at the peptide–spectrum matches (PSMs), peptide, protein, and glycopeptides levels. For proteomics, the quantitation of peptides across PMM2-CDG and control fibroblasts was performed using TMT reporter ion intensities using “reporter ion quantifier” node. A minimum of one unique peptide was considered for the reliable identification of proteins. The reporter ions’ intensity was obtained using an integration tolerance of 30 ppm around the reporter masses. For reliable quantitation, only PSMs filtered for a co-isolation threshold of 70% and an average S/N value of 10 were considered for protein quantitation. To quantify glycopeptides, reporter ion quantification was performed for glycoproteomics raw files in Proteome Discoverer version 2.5 and glycopeptide IDs obtained from pGlyco 3 were matched with quantitation on a scan-to-scan basis (MS/MS). The data were searched against the annotated gene list for autophagy, mitophagy and endoplasmic reticulum-associated degradation (ERAD), which was generated from Gene Ontology (GO) resources [38]. For mitochondrial proteins, annotated gene list was generated from MitoCarta 3.0 [38].

Our proteomics and glycoproteomic pipelines, unfortunately, have limited the maximal number of individuals (*n* = 6), we could include in these analyses. As a result, we have used 3 PMM2-CDG and 3 control fibroblast lines. These cell lines have been well-established models in our laboratory. In addition, clinical and genetic data, as well as fibroblast characteristics, have been well-documented for these cell lines. We have also implemented rigorous controls to avoid any additional variability due to experimental conditions. We acknowledge this as a potential limitation of our study.

### 2.9. Statistical Analyses

All data are expressed as mean ± SD. All statistical analyses were performed using GraphPad Prism version 8.3 or 9.0. Statistical analyses for both proteomics and glycoproteomics data were performed using the publicly available computational platform Perseus [39] and R studio. Two sample *t*-tests were used to compare two groups. Correlation coefficients were calculated using Pearson’s r (*p* < 0.05).

## 3. Results

### 3.1. Clinical Data

In the cohort of seven male and two female patients, the mean age was 6 years (range = 2 to 13 years). The total median NPCRS score of patients was 21 (moderate phenotype), ranging from 7 to 30: one had mild, six had moderate, and two had severe phenotypes. The most common *PMM2* variant was c.422G > A; p.R141H found in seven out of nine patients. Sorbitol level was abnormal in three out of seven patients, and mannitol was abnormal in two out of seven patients. Lactic acid measured in blood was normal in all the patients, except in P5, where it was slightly elevated. All the patients had decreased PMM activity (average 136 nmol/h/mg Protein; normal control ≥ 700; Table 1).

We also assessed PMM enzyme activity in PMM2-deficient fibroblasts. We found a 5.15-fold lower abundance of PMM enzyme activity in fibroblasts of individuals with PMM2-CDG compared to the controls (average 142 nmol/h/mg protein in PMM2-deficient fibroblasts; normal control ≥ 700).

### 3.2. Autophagy in PMM2-CDG Derived Fibroblasts

We found a twofold increase in the autophagy marker LC3-II/ in P1 and P2 compared to the controls (Figure 1). Interestingly, three patients (P7, P8 and P9) presented with decreased LC3-II abundance compared to the controls. Quantification of LC3-II indicated no significant difference in autophagosome marker LC3-1 and LC3-II in PMM2-CDG cell lines (*n* = 8) compared to healthy controls (*n* = 3) measured via Western blot analysis (Figure 1). A correlation analysis between LC3II abundance and CDG severity scores (NPCRS) indicated an inverse association between the PMM2-CDG severity score and the autophagy marker LC3-II. Lower levels of the autophagy marker LC3-II were associated with a more severe clinical presentation, i.e., higher NPCRS scores. In contrast, high levels of LC3-II autophagy marker predicted less severe clinical presentations as assessed via NPCRS (Figure 1). In summary, LC3II autophagy markers, although they varied significantly among individuals with PMM2-CDG, were inversely correlated with the PMM2-CDG severity scores.

### 3.3. Bioenergetics of PMM2-CDG Fibroblasts

#### 3.3.1. Cell Mito Stress Test

First, we assessed mitochondrial function using the Seahorse Mito Stress test as well as assessed the extracellular acidification rate. Analysis of mitochondrial oxygen consumption of fibroblasts of eight PMM2 patients and three controls revealed a 35% reduction in maximal respiration (*p* < 0.03) and a 26% decrease in ATP-linked respiration (*p* < 0.04) in PMM2-CDG patients compared to controls (Figure 2, Appendix A). OCR data were normalized to citrate synthase activity that was comparable between PMM2-CDG and control individuals.

Next, we assessed if PMM2-CDG fibroblasts presented with a functional decline in cellular ATP production using the Seahorse XF^®^ ATP rate assay. We found that the glycolytic ATP production rate (glycoATP Production Rate) was significantly decreased (30%, *p* = 0.04) in PMM2-CDG compared to controls (Figure 3). Similarly, the total ATP production rate was also lower (21%) in PMM2-CDG compared to controls (*p* < 0.02). Mitochondrial ATP production (mitoATP Production Rate) was comparable between PMM2-CDG and controls (Figure 3). Additional parameters, such as XF ATP rate index, % of glycolysis and % of oxidative phosphorylation, were also comparable between groups (Appendix A).

#### 3.3.2. Mitochondrial Electron Transport Chain (mtETC) Complex Enzymology

The activities of mtETC complex I (CI), complex II (CII), complex III (CIII) and complex IV (CIV), as well as citrate synthase (CS) activity and protein concentration, were assayed in homogenates from cultured cell lines derived from seven PMM2 patients and three controls. CS activity is a common proxy for mitochondrial mass, as CS is ubiquitously expressed at high levels in the mitochondrial matrix. Alternatively, protein concentration is a common proxy for cell number. CI and CIV activity were normalized to both CS activity and protein concentration, as well as to cell number. PMM2-CDG cell lines displayed a reduced CI activity relative to CS activity (−58%, *p <* 0.03, Figure 4A), though not relative to protein concentration or cell number (Figure 4 B,C). No difference in CII, CIII and CIV activity was observed, regardless of normalization (Figure 4D–F depicts CIV activity). Notably, PMM2-CDG cells also displayed increased activity of CS relative to cell number (Figure 4H), though not relative to protein concentration (Figure 4G), which may account for the disparity between variations in CI activity between PMM2-CDG cell lines and controls depending on normalization. An increased CS activity relative to cell number is consistent with an increase in mitochondrial mass in the PMM2 cell lines and may indicate a compensatory increase in mitochondrial biogenesis in response to reduced ATP production. CI activity normalized to CS activity was significantly correlated (R^2^ > 0.70) with basal respiration as measured via Seahorse, while CI activity normalized to protein concentration was significantly correlated with ATP-linked respiration (Figure 4I).

### 3.4. Phospho-Ubiquitin p-S65-Ub, a Specific Marker for Mitochondrial Stress

We used a previously established sensitive ELISA assay as a readout for phosphor–ubiquitin (p-S65-Ub) that is PINK1-and PRKN-dependently generated upon mitochondrial damage and serves as a specific marker for mitophagy induction [34,40,41]. Using lysates from eight PMM2 fibroblast lines, we found a suggestive correlation of p-S65-Ub levels with *PMM2* enzyme activity levels and a strong correlation with Sorbitol levels (Figure 5A). There was no correlation between p-S65-Ub and the autophagosome marker LC3 or other assessed parameters (Figure 5B) 

### 3.5. PMM2-CDG Fibroblasts Demonstrate Altered Expression of Proteins Involved in Autophagy, Mitophagy, Endoplasmic Reticulum Associated Degradation and Mitochondria

Multiplexed proteomics study of fibroblasts was performed in three PMM2-CDG and three control fibroblasts to evaluate the expression of different proteins, which are annotated in autophagy, endoplasmic-reticulum-associated degradation (ERAD), mitochondria and mitophagy. Multiplexed proteomics study identified a total of 8368 proteins and 127,484 peptides. Of these, we identified and quantified a total of 301, 226, 80 and 877 proteins related to autophagy, mitophagy, ERAD and mitochondrial proteins, respectively. The expression pattern of these proteins with their significance values for each pathway is represented in volcano plots (Figure 6A–D). First, we assessed whether PMM2 protein abundance was indeed decreased in PMM2 deficient fibroblast compared to controls. In line with our previous results [42], we found that the abundance of PMM2 protein was reduced by 20% in PMM2-deficient fibroblasts compared with controls, although this change was not significant. We found that the upregulation of several proteins associated with autophagy and mitophagy in patient fibroblasts. The top significantly (*p*-value < 0.05) upregulated autophagy-related proteins (average FC > 1.3) included death-associated protein 1 (DAP), mitochondria-eating protein (SPATA18), platelet-activating factor acetylhydrolase IB subunit alpha2 (PAFAH1B2), malate dehydrogenase, cytoplasmic (MDH1), endophilin-B1 (SH3GLB1), vacuolar protein-sorting-associated protein 25 (VPS25), ubiquitin-like-conjugating enzyme ATG3 (ATG3) and vacuolar protein sorting-associated protein 4B (VPS4B) (Figure 6E). The top significantly (*p*-value < 0.05) upregulated mitophagy-related proteins (average FC > 1.3) included forkhead box protein O3 (FOXO3), tubulin α-1C chain (TUBA1C), ubiquitin-conjugating enzyme E2 L3 (UBE2L3) and ubiquitin-conjugating enzyme E2 N (UBE2N), whereas leucine-rich PPR motif-containing protein (LRPPRC) and polycystin-2 (PKD2) were downregulated in patients (Figure 6F). The expression of the majority of ERAD-related proteins was comparable between patients and controls. Only four proteins that were significantly changed and mildly upregulated in patients included UBX domain-containing protein 6 (UBXN6), ataxin-3 (ATXN3), nuclear protein localization protein 4 homolog (NPLOC4), ubiquitin recognition factor in ER-associated degradation protein 1 (UFD1) (Figure 6G). The upregulation of autophagy and mitophagy proteins in patients indicates the cellular stress in PMM2-CDG patients. These data also indicate the slight increase in unfolded protein response because of the defect in N-glycosylation biosynthesis machinery in patients.

We also evaluated the expression of mitochondrial proteins in patients. Notably, out of identified 877 mitochondrial proteins, 34 proteins were significantly upregulated, and 82 proteins were significantly downregulated with *p* < 0.05. The top upregulated proteins included peroxiredoxin-6 (PRDX6), quinone oxidoreductase (CRYZ), protein NipSnap homolog 3A (NIPSNAP3A) and sterol 26-hydroxylase (CYP27A1). The top downregulated proteins included transmembrane protein 65 (TMEM65), coiled–coil–helix–coiled–coil–helix domain-containing protein 2 (CHCHD2), protein FAM210B and rRNA methyltransferase 3 (MRM3) (Figure 6H). Interestingly, we also observed the downregulation in several subunits from NADH dehydrogenase complex and mitochondrial ribosomal subunits in PMM2-CDG. The downregulation in several mitochondrial proteins, including complex I and mito ribosomes, confirms the stress on mitochondria in patients. A complete list of significantly changing proteins for each pathway interrogated with the fold-change can be found as Appendix A.

### 3.6. PMM2-CDG Fibroblasts Exhibit Distinct Glycosylation Pattern in Autophagy Proteins

We could confirm the abundances of 4356 individual intact glycopeptides with 282 unique glycan compositions on 904 glycosylation sites of 500 glycoproteins. In our quantitative N-glycoproteomic analysis, we identified 271 glycopeptides which are derived from 12 autophagy-related proteins. We observed reduced N-glycosylation in PMM2-CDG fibroblasts on different sites originating from these proteins. These included diverse glycan moieties ranging from high mannose and complex/hybrid to sialylated and fucosylated glycans. A differential chord diagram quantitatively visualizes glycosylation differences in patients as compared to control fibroblasts on distinct glycosylation sites of various proteins (Figure 7A).

The chord diagram also shows the micro and macro heterogeneity of glycan compositions on different autophagy proteins in fibroblasts. Interestingly, we found significant downregulation in 20 glycopeptides that were derived from lysosome-associated membrane glycoprotein 1; LAMP1 (six glycopeptides), lysosome-associated membrane glycoprotein 2; LAMP2 (five glycopeptides), hyaluronan and proteoglycan link protein 1; HAPLN1 (seven glycopeptides) and carbohydrate sulfotransferase; CHST3 (one glycopeptide). Interestingly, the expression of these proteins at the protein level was not significantly altered in our proteomics data. The heatmap of these glycopeptides with sites and glycan composition is shown in (Figure 7B). A complete list of significantly changing glycoproteins for each pathway interrogated with the fold-change can be found in Appendix A.

## 4. Discussion

In this research study, we aimed to investigate the hypothesis that PMM2-CDG is associated with secondary mitochondrial dysfunction and impaired autophagy/mitophagy in patient-derived fibroblasts with differing genotypes. Our findings revealed that although the LC3II autophagy marker in patient fibroblasts exhibited significant variation among individuals with PMM2-CDG, its levels were inversely correlated with the clinical CDG severity score and urine sorbitol levels. Additionally, we observed substantial changes not only in abundance but also in the glycosylation of several proteins involved in the autophagy/mitophagy pathways. These results suggest that both mitochondrial function and autophagy/mitophagy could potentially influence the clinical presentation of PMM2-CDG.

Secondary suboptimal mitochondrial function has been reported in individuals with distinct types of CDG [4,7,9,43]. Our results provide further evidence that cellular bioenergetics is impaired in PMM2-CDG. Specifically, mitochondrial oxygen consumption was reduced during maximal respiration and ATP-linked respiration in fibroblasts with PMM2-CDG compared to controls (Figure 2 and Figure 3). Secondary suboptimal mitochondrial function in PMM2-CDG fibroblasts has been further corroborated by reduced CI activity (Figure 4A). Interestingly, CI activity was correlated with mitochondrial oxygen consumption during basal and ATP-linked respiration (Figure 4C). In a recent study, Himmelreich et al. demonstrated that a glycosylation defect was associated with changes in several metabolites strongly associated with mitochondrial (dys)function, such as acylcarnitines, amino acids, lysosomal enzymes, and lipids [43].

p-S65-Ub is the joint product of the ubiquitin kinase PINK1 and the E3 ligase PRKN. p-S65-Ub chains are built onto the surface of damaged mitochondria as a specific degradation signal [40,44,45]. Such labeled mitochondria are then recognized by autophagy adapters, engulfed in autolysosomes, and eventually degraded via autophagy in lysosomes [45]. In this study, p-S65-Ub levels of the analyzed fibroblasts correlated with two CDG-associated factors, PMM2 enzyme activity and sorbitol (Figure 5B). This suggests disease-associated mitophagy changes in PMM2-CDG. As we have previously identified sorbitol as a biomarker for the clinical severity of PMM2-CDG [42], the strong correlation of sorbitol with pS65-Ub may indicate that mitophagy changes are closely linked to severe forms of CDG, further supporting the hypothesis of secondary suboptimal mitochondrial function. It is possible that the observed secondary suboptimal mitochondrial function is caused by downstream impairment of the autophagic–lysosomal systems triggered by the buildup of misfolded proteins due to primary glycosylation deficits.

Proteomics data showed increased expression of autophagy and mitophagy-related proteins in all three PMM2-CDG patients, suggesting that the defect in N-glycan synthesis could lead to cellular stress and, eventually, cell death (Figure 6). Death-Associated Protein Kinase 1 (DAPK1) is a known apoptosis and autophagy regulator that acts as a critical component in the ER stress-induced cell death pathway [46]. The overexpression of death-associated protein 1 could also be linked with increased cell death in PMM2-CDG fibroblasts. Another autophagy protein, endophilin B1 (Bax-Interacting Factor 1 or SH3GLB1), is involved in the early stage of autophagy and is known to interact with Beclin1 through UVRAG (ultraviolet irradiation resistant-associated gene) to regulate the activation of vacuolar protein-sorting-associated protein, and eventually, initiate autophagy [47,48]. It is also shown that endophilin B1 foci co-localize with microtubule-associated protein light chain 3 (LC3) [47]. The overexpression of all these proteins was observed in PMM2-CDG fibroblasts (Figure 6). We also observed a mild increase in specific proteins involved in endoplasmic-reticulum-associated degradation (ERAD). ERAD is a known quality control system in the secretory pathway and is responsible for degrading misfolded proteins, including those with glycosylation defects. Two topmost abundant proteins in PMM2-CDG, UBX domain-containing proteins and attaxin-3, bind with valosin-containing protein (VCP/p97) and help in the retrotranslation of unfolded proteins into the cytosol from the ER [49,50]. A reduced abundance of several mitochondrial proteins, including several subunits of the respiratory complex I (NADH ubiquinone oxidoreductase) and mito ribosomal subunits, suggest the mitochondrial stress in patient fibroblasts. The reduced abundance of several subunits of NADH ubiquinone oxidoreductase corroborates our observation of reduced respiratory complex I activity in PMM2-CDG fibroblasts.

Autophagy regulatory proteins and core machinery are targets of glycosylation [51]. The function of the ATG proteins, in particular their ability to interact with a number of macroautophagic regulators, is modulated by posttranslational modifications (PTMs) such as phosphorylation, glycosylation, ubiquitination, acetylation, lipidation, and proteolysis [52,53]. In line with this notion, here we report on PMM2-CDG fibroblasts exhibiting altered glycosylation patterns in several autophagy proteins. Our recent study had shown that the defect in the *PMM2* gene cause global hypoglycosylation on several cellular proteins in patient-derived fibroblasts, which included autophagy-related proteins, lysosome-associated membrane proteins 1 (LAMP1) and LAMP2 [42]. In our current study, we confirmed the hypoglycosylation of LAMP1 and LAMP2 in PMM2-CDG fibroblasts. Besides LAMP1 and LAMP2, other significantly hypoglycosylated proteins included hyaluronan and proteoglycan link protein 1 (HAPLN1) and carbohydrate sulfotransferase 3 (CHST3). Both HAPLN1 and CHST3 are the major components of the extracellular matrix and are involved in cell migration and differentiation [54,55]. Hypoglycosylation of these proteins might reduce their functions that could affect cell growth, and eventually promote cell death.

## 5. Conclusions

Autophagy, a process of cellular self-eating, may initially appear puzzling, but its significance becomes clearer when understood as a single cell’s response to impaired cellular bioenergetics, particularly suboptimal mitochondrial function in PMM2-CDG. Essentially, autophagy serves as a critical housekeeping pro-survival pathway, adapting to challenging conditions [22,23,24]. This notion gains support from our research, indicating that increased autophagy correlates with improved clinical outcomes in individuals with PMM2-CDG (Figure 1B and Figure 5B). Consequently, exploring the potential to selectively activate or deactivate autophagy gene-dependent survival pathways or manipulate mitochondrial function may offer additive therapeutic benefits when combined with existing drugs for PMM2-CDG treatment [42,56].

## Figures and Tables

**Figure 1 genes-14-01585-f001:**
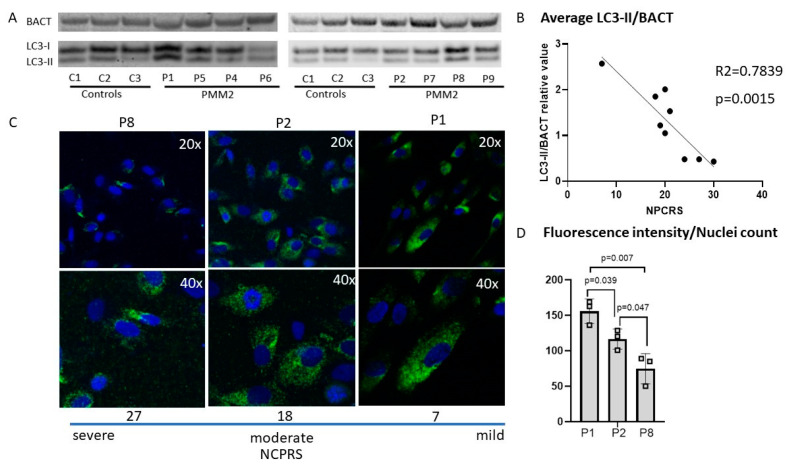
The CDG severity score is inversely correlated with autophagosome marker LC3 measured by Western blot and immunocytochemistry. (**A**). Western blot analysis including an inverse correlation between LC3-II/BACT and CDG severity score—NPCRS was found in PMM2-deficient patients’ fibroblasts. (**B**). Higher NPCRS scores (more severe CDG phenotypes) are linked to lower LC3-II/BACT values. (**C**). Immunocytochemistry with LC3 staining was used as a confirmatory test and further analysis of PMM2-CDG cell lines with severe, moderate, and mild phenotype. (**D**). Quantification of immunostaining indicated the association between the CDG phenotype severity and the LC3 intensity. P1 presents with a mild CDG score (7), P2 presents with a moderate CDG score (18), and P8 presents with a severe CDG score. Patient (P8) with severe CDG phenotype showed lower levels of LC3 staining compared to a patient (P1) with a mild CDG score. DAPI nucleus staining (blue) and LC3 staining (green).

**Figure 2 genes-14-01585-f002:**
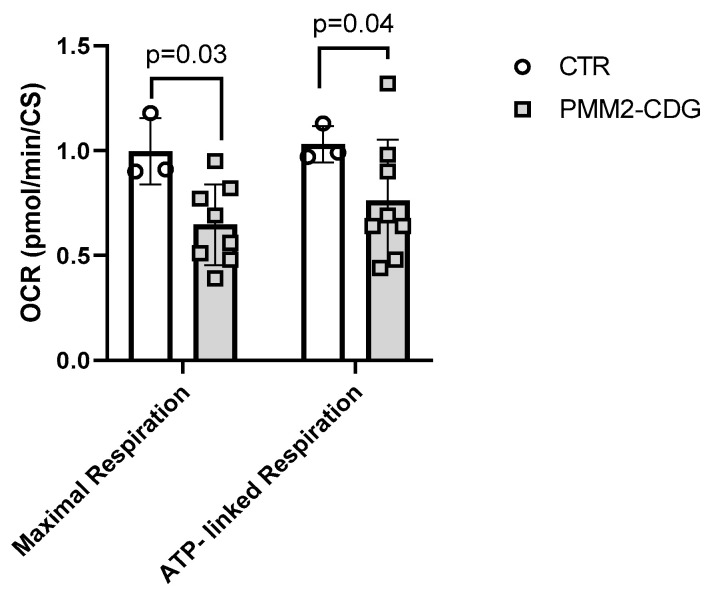
Mitochondrial oxygen consumption. Seahorse XF^®^ respirometry was used to assess mitochondrial respiration in control and patient fibroblasts. PMM2-CDG patients (*n* = 8) showed a 35% reduction in maximal respiration (*p* < 0.03) and a 26% decrease in ATP-linked respiration (*p* < 0.04). Oxygen consumption rates were normalized to citrate synthase activity. Data are indicated as mean ± SD. FCCP, carbonyl cyanide phenylhydrazone; OCR, oxygen consumption rate.

**Figure 3 genes-14-01585-f003:**
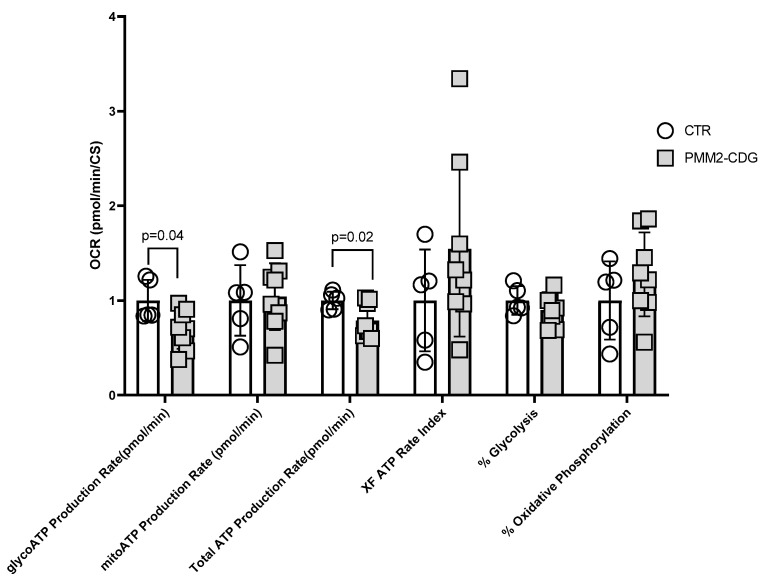
Mitochondrial and non-mitochondrial ATP production of PMM2-CDG and control fibroblasts. Seahorse XF^®^ real-time ATP rate assay revealed that glycoATP production and total ATP production rates were significantly decreased in PMM2-CDG patients’ fibroblasts (*n* = 8) compared to healthy controls (*n* = 5).

**Figure 4 genes-14-01585-f004:**
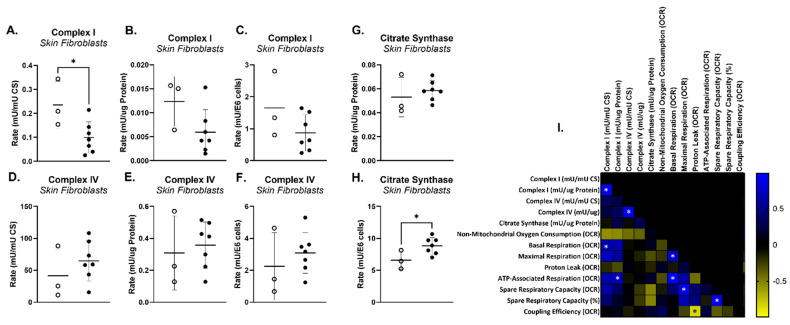
Mitochondrial electron transport chain (mtETC) complex enzymology. (**A**–**H**) Complex I and complex IV activities in 7 PMM2 skin fibroblast cell lines and 3 control skin fibroblast cell lines normalized to citrate synthase (CS) activity, protein concentration, and cell number. (**I**). Pearson correlation between mtETC complex enzymology and Seahorse oxygen consumption rates (OCR).

**Figure 5 genes-14-01585-f005:**
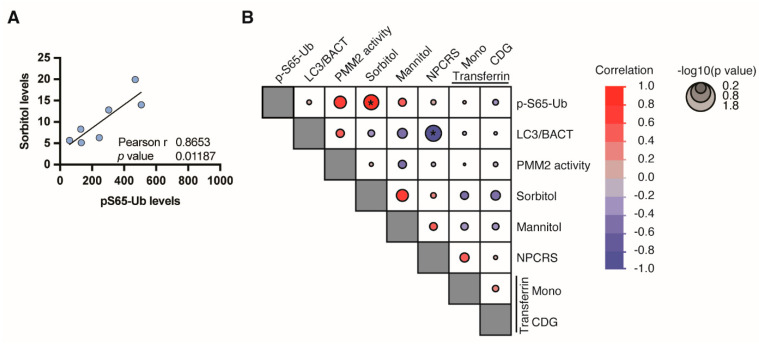
p-S65-Ub as a marker of PINK1/PRKN-dependent mitophagy. P-S65-Ub levels were determined by ELISA. (**A**) Pearson’s correlation of p-S65-Ub with Sorbitol levels. (**B**) Correlation matrix of all assessed parameters. Negative correlations are shown in red, positive in blue gradient. Size of the circle indicates significance, with larger circles having a lower *p* value (*, *p* < 0.05).

**Figure 6 genes-14-01585-f006:**
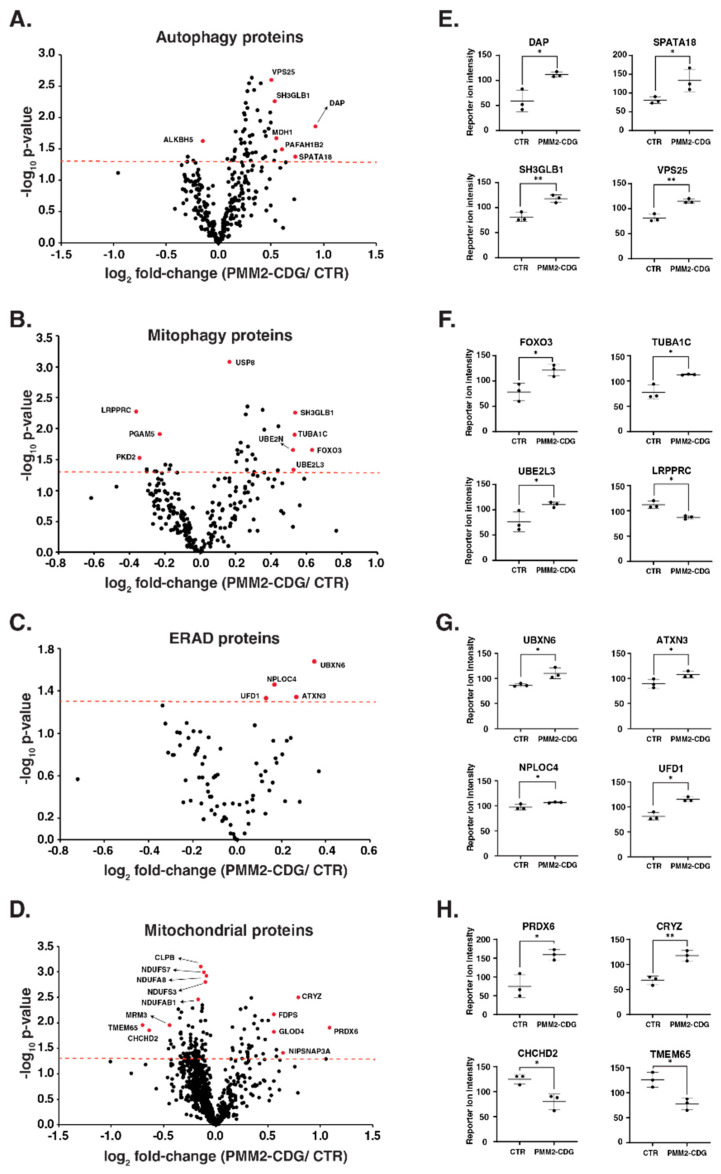
Proteomic changes in patient-derived PMM2-deficient fibroblasts. Volcano plots depicting the differentially abundant proteins in patient-derived PMM2-deficient fibroblasts for the proteins related to autophagy (**A**), mitophagy (**B**), endoplasmic-reticulum-associated degradation (ERAD) (**C**) and mitochondria (**D**). X-axis is log_2_ fold change (PMM2-CDG/controls), and Y-axis is the negative logarithm of *p*-value of t test for significance. The horizontal dashed red line represents the cutoff for significance (*p* < 0.05). Some of the highly changing are marked in red circles, and protein names are provided. Dot plots showing reporter ion intensities are represented in (**E**–**H**) for top-changing proteins related to autophagy, mitophagy, ERAD and mitochondria, respectively. Y-axis is the reporter ion intensity of TMT channels. Each dot in the plot represents the individual control or patient sample. *p* < 0.05 (*) and *p* < 0.01 (**).

**Figure 7 genes-14-01585-f007:**
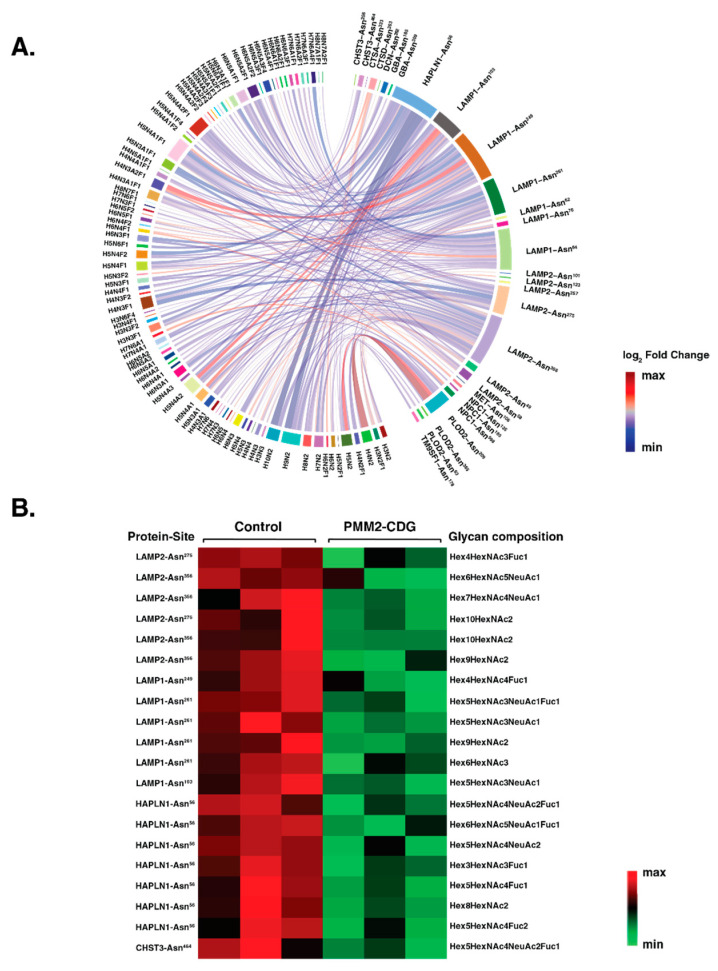
Site-specific glycosylation changes in autophagy-related proteins. (**A**) Differential chord diagram depicting the site-specific glycosylation changes for autophagy-related proteins in PMM2-CDG as compared to control fibroblasts. Proteins with different glycosylation sites are indexed on the right of the diagram and connected via chords to respective identified glycan structures on the left. The fold-change pattern is color coded. (**B**) Heatmap of significantly changing glycopeptides (*p* < 0.05) with protein names (LAMP2; lysosome-associated membrane glycoprotein 2, LAMP1; lysosome-associated membrane glycoprotein 1, HAPLN1; hyaluronan and proteoglycan link protein 1 and CHST3; carbohydrate sulfotransferase 3) sites and glycan composition. The pattern is color coded.

**Table 1 genes-14-01585-t001:** Overview of PMM2 activity, polyol levels (sorbitol, mannitol), lactate levels and clinical phenotype characteristics in PMM2-CDG patients.

ID	PMM2 Enzyme Activity ≥ 700 CT (nmol/h/mg Protein)	Sorbitol Level < 10 μmol/mmol CT	Mannitol Level < 15 μmol/mmol CT	Lactate Level	NPCRS Total (Condition)	Transferrin	
						Mono-Oligo/Di-Oligo Ratio ≤ 0.06	CDG A-Oligo/Di-Oligo Ratio ≤ 0.011
P1	135	8.33	10.2	normal	7 (mild)	0.0260	0.0122
P2	429	-	-	normal	20 (moderate)	↑0.4908	↑0.2204
P3	78	6.32	0.81	normal	18 (moderate)	↑0.69	↑0.196
P4	117	↑14	9.52	normal	21 (moderate)	↑0.58	↑0.170
P5	58	5.16	6.96	abnormal	19 (moderate)	↑0.35	↑0.76
P6	228	↑12.8	14	-(abnormal in a setting of meningitis)	20 (moderate)	↑0.242	↑0.058
P7	44	↑19.93	↑648.6	normal	24 (moderate)	↑0.1	0.007
P8	63	5.67	↑121.7	normal	27 (severe)	↑0.65	↑0.194
P9	68	-	-	normal	30 (severe)	↑0.086	↑0.042

## Data Availability

The data presented in this study are available upon request from the corresponding author. The data are not publicly available due to restrictions on patient privacy.

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
