# Peer review of "Interplay of Impaired Cellular Bioenergetics and Autophagy in PMM2-CDG"

_genes, 2023, doi:10.3390/genes14081585_

Round 1
Reviewer 1 Report
Dear authors, the article you presented is interesting and provides some novel investigational aspects regarding CDC. However, i would suggest some modifications to improve its quality at current stage.
Below are some indications for improving the manuscript:
- introduction: add specification for PMM2 gene (protein product and function)
- results: Fig1. Use same dimensions for blot panels and align them.
in general the authors to improve the quality of figures. in some cases the labels are not aligned and graphs appear confused. I would recommend to increase the size of the labels (including protein names) because I was not able to read the majority of them in the low-res pdf file I have downloaded.
- Have the author assessed whether PMM2 protein is normally expressed in their samples? or is it a known information from literature that PMM2 variants do not affect the expression, or the contrary? This should be checked in the proteome of these cells, because this could be a useful hint to detect some co-isolation issues with TMT labelling.
- Proteomics: besides the specification of autophagy, ERAD, mitochondria and mitophagy proteins, one thing that I have probably missed to read is the detail about the total number of proteins identified. With an Orbitrap Eclipse and the high pH fractionation method I would expect to see at least 3000-4000 proteins identified from a cell lysate. Some other details are missing, such as the threshold used for protein identification, number of unique peptides, number of quantitative values accepted to retain proteins before missing values imputation, etc.
- Glycoproteomics: as above, the authors should provide more details about proteins and peptides identified after glycan-enrichment. This is important to understand the difference (also in qualitative terms) of glycosylated peptides with respect to control cells. I understood from the heat map that glycopeptides for some of autophagy proteins are down-represented. However, the pattern for all the other proteins are not shown.
- For both proteomic and glycoproteomic analyses, the authors should submit their .raw file to some repositories (e.g. PRIDE).
Author Response
Ad Reviewer 1
We thank the reviewer for their thoughtful comments and suggestions. We have revised our manuscript in line with these suggestions.
Reviewer: Introduction: add specification for PMM2 gene (protein product and function)
Authors: This information has been added to the Introduction
Reviewer: Results: Fig1. Use same dimensions for blot panels and align them.
Authors: We have updated figure 1 in line with the reviewer’s suggestion.
Reviewer: In general, the authors to improve the quality of figures. in some cases, the labels are not aligned, and graphs appear confused. I would recommend increasing the size of the labels (including protein names) because I was not able to read the majority of them in the low-res pdf file I have downloaded.
Authors: We have revised all figures and increased resolution, so figure labels can be read. We respectfully ask the reviewer to download the high-resolution images to check for figure quality in the revised manuscript.
Reviewer: Have the author assessed whether PMM2 protein is normally expressed in their samples? or is it a known information from literature that PMM2 variants do not affect the expression, or the contrary? This should be checked in the proteome of these cells, because this could be a useful hint to detect some co-isolation issues with TMT labelling.
Authors: In this study, PMM2 protein was indeed present, and its expression was reduced by around 20% in PMM2 protein abundance in PMM2-deficient fibroblasts compared with controls, although this change was not significant. The PMM enzyme level was reduced by 4.93-fold change (average 142 nmol/h/mg Protein; normal control ≥700). We have now included this information in the revised manuscript.
Reviewer: Proteomics: besides the specification of autophagy, ERAD, mitochondria and mitophagy proteins, one thing that I have probably missed to read is the detail about the total number of proteins identified. With an Orbitrap Eclipse and the high pH fractionation method I would expect to see at least 3000-4000 proteins identified from a cell lysate. Some other details are missing, such as the threshold used for protein identification, number of unique peptides, number of quantitative values accepted to retain proteins before missing values imputation, etc.
Authors: We have added the requested information to the relevant methods and results section.
Reviewer: Glycoproteomics: as above, the authors should provide more details about proteins and peptides identified after glycan-enrichment. This is important to understand the difference (also in qualitative terms) of glycosylated peptides with respect to control cells. I understood from the heat map that glycopeptides for some of autophagy proteins are down-represented. However, the pattern for all the other proteins is not shown.
Authors: We have added the requested information to the relevant methods and results section.
Reviewer: For both proteomic and glycoproteomic analyses, the authors should submit their .raw file to some repositories (e.g. PRIDE).
Authors: The co-author responsible for the proteomic and glycoproteomic analysis is at the moment in India. He tried to upload the data, but the VPN connection was week. We respectfully ask the reviewer and the editors to allow us adding this information later (either in another round of revisions and at proofreading). Thanks for your understanding.

Reviewer 2 Report
This is an interesting study, however they authors need to go further to fully support the conclusions they have drawn. There are several issues (outlined below) which need to be addressed before this manuscript can be considered for publication.
The interpretation of the LC3 western blots not convincing. It well established that LC3-II tends to be much more sensitive in immunoblotting than LC3-I. As a result simply by measuring the apparent amount of LC3-II on blots is not a very robust measure of autophagy. Furthermore, it is not clear the relationship between C samples and P samples – there is often variation in LC3 levels detected by western blotting between different cultures from the same cell line let alone different patients. With this in mind the data presented did not convince me that they support the conclusions that had been drawn.
All the data points for Figure 1D need to be shown as well as bars (as the authors have done in figure 2).
Traces from the seahorse analyser for the data plotted in figure 2 need to be included in the supplementary data.
It is not clear how the authors have calculated their significant differences as the data points and error bars are overlapping in Figure 3.
Could the authors also make it clearer which samples have been tested to generated these data – up until this point in the paper there have been 3 control samples and 8 PPM2-CDG samples. Now there are 5 controls and 7PPM2-CDG samples – how does this relate the previous data?
Authors have not referred to panels 4G, H or I in the text. Figures 4G &H seems to suggest based on text that the PPM2 samples have significantly more mitochondrial mass per cells? If this is correct surely this should be highlighted and commented upon in the text?
Additionally, there is no details in the figure legend or in the text as to panel 4I to enable the reader to understand what it shows or why it is there.
The ELISA data used to generate Figure 5 needs to be included in the supplementary data file - there is no actual data presented currently to convince the reader that these correlations are accurate.
Proteomic studies – which three PPM2 cell lines were used and why those specific ones chosen. All the data presented up to this point in the paper has shown that there is a lot of variation between the cell lines – therefore the choice of cell line will have impacted on data generated and the ability to draw any conclusions from the proteomic studies.
Typo 426 – dat should be data
The quality of the figures 6 & 7 is too low, due to the size to read the figure you need to zoom in, and when you do this they are all pixelate and still unreadable – this needs to be corrected. Currently due to the low quality figures it is hard to evaluate what they are showing and whether they support the conclusions being drawn.
The authors are able to address these significant issues with their manuscript before it can be properly evaluated.
Overall the English in this manuscript is fine, there are minor typos and grammatical errors which need to be addressed.
Author Response
Ad Reviewer 2
Reviewer: This is an interesting study, however they authors need to go further to fully support the conclusions they have drawn. There are several issues (outlined below) which need to be addressed before this manuscript can be considered for publication.
Authors: We thank the reviewer for their comments and critique. We have carefully addressed all these comments and critiques.
Reviewer: The interpretation of the LC3 western blots not convincing. It well established that LC3-II tends to be much more sensitive in immunoblotting than LC3-I. As a result, simply by measuring the apparent amount of LC3-II on blots is not a very robust measure of autophagy. Furthermore, it is not clear the relationship between C samples and P samples – there is often variation in LC3 levels detected by western blotting between different cultures from the same cell line let alone different patients. With this in mind the data presented did not convince me that they support the conclusions that had been drawn.
Authors: We thank the reviewer for their comment. We agree that LC3II is more sensitive in Western blotting that LC3I. Therefore, we focus our analysis to assess relative LC3II levels normalized to beta actin levels in controls and PMM2-CDG individuals. Although LC3II levels varied between individuals (both controls and PMM2-CDG individuals), and therefore LC3II levels were not significantly different between controls and patients, importantly, we found a strong inverse correlation of relative LC3II levels and clinical severity as assessed by the Nijmegen Pediatric Clinical Research Score (NPCRS). We have now clarified this in the revised manuscript.
Reviewer: All the data points for Figure 1D need to be shown as well as bars.
Authors: We have updated the Figure 1D in line with the reviewer’s comment.
Reviewer: Traces from the seahorse analyser for the data plotted in figure 2 need to be included in the supplementary data.
Authors: Traces from the seahorse analyser for the data plotted in figure 2 have been now included in supplementary Table 1.
Reviewer: It is not clear how the authors have calculated their significant differences as the data points and error bars are overlapping in Figure 3. Could the authors also make it clearer which samples have been tested to generate these data – up until this point in the paper there have been 3 control samples and 8 PPM2-CDG samples. Now there are 5 controls and 7PPM2-CDG samples – how does this relate the previous data?
Authors: We appreciate the reviewer's feedback and would like to address their concerns. The significance of the differences was calculated using unpaired T-test. As for the number of control and PMM2-CDG samples used in the study, we acknowledge that the number of individuals with PMM2-CDG in both Figure 2 and 3 is 8, however, in Figure 3 included 5 controls. In Figure 2, unfortunately, we could only include three controls. The number of available control fibroblasts in our biobank varied at the time of experiments depicted in Figure 2 and 3. We also acknowledge that three of the patients presented with comparable seahorse analyzer data to controls. Therefore, some data points and error bars overlap with controls in Figure 3. However, our statistical analysis revealed significant differences between PMM2-CDG individuals and controls.
We have also added individual data plotted in Figure 3 to Supplementary Table 1.
Reviewer: Authors have not referred to panels 4G, H or I in the text. Figures 4G &H seems to suggest based on text that the PPM2 samples have significantly more mitochondrial mass per cells? If this is correct surely this should be highlighted and commented upon in the text?
Authors: We thank the reviewer for their comment. We routinely use citrate synthase activity to normalize mitochondrial complex activities as citrate synthase activity indeed could correlate with mitochondrial mass. We also assess the relative activity of citrate synthase normalized to protein as well as to cell number. PMM2-CDG fibroblasts presented with increased citrate synthase activity when normalized to cell number, but this was comparable between PMM2-CDG and control fibroblasts when normalized to protein content. Increased citrate synthase activity normalized to cell number could indeed mean increased mitochondrial mass. It could however, also mean that PMM2-CDG fibroblasts are smaller, delayed in their growth compared to controls. We would only consider increased mitochondrial mass, if citrate synthase activity were increased normalized to both protein and cell number. For these reasons, we do not feel confident to suggest that mitochondrial mass is increased in PMM2-CDG fibroblast.
Reviewer: Additionally, there is no details in the figure legend or in the text as to panel 4I to enable the reader to understand what it shows or why it is there.
Authors: We have added details about panel 4I to the figure legend and referenced Fig4I in the main text.
Reviewer: The ELISA data used to generate Figure 5 needs to be included in the supplementary data file - there is no actual data presented currently to convince the reader that these correlations are accurate.
Authors: individual ELISA data points plotted in Figure 5, are mow included in Supplementary Table 3.
Reviewer: Proteomic studies – which three PPM2 cell lines were used and why those specific ones chosen. All the data presented up to this point in the paper has shown that there is a lot of variation between the cell lines – therefore the choice of cell line will have impacted on data generated and the ability to draw any conclusions from the proteomic studies.
Authors: Our proteomics and glycoproteomic pipelines unfortunately have limited the maximal number of individuals (n=6), we could include in these analyses. As a result, we have used 3 PMM2-CDG and 3 control fibroblast lines. These cell lines have been well-established models in our laboratory. In addition, clinical and genetic data as well as fibroblast characteristics have been well-documented for these cell lines. We have also implemented rigorous controls to avoid any additional variability due to experimental conditions. We acknowledge this as a potential limitation of our study.
Reviewer: Typo 426 – dat should be data
Authors: This has been corrected
Reviewer: The quality of the figures 6 & 7 is too low, due to the size to read the figure you need to zoom in, and when you do this, they are all pixelate and still unreadable – this needs to be corrected. Currently due to the low-quality figures it is hard to evaluate what they are showing and whether they support the conclusions being drawn.
Authors: We have revised Figure 6 and 7, improved their resolution to accommodate the reviewer’s comment.

Round 2
Reviewer 1 Report
Dear Authors,
thanks for taking into account all the requested modifications.
I do suggest that the manuscript can be accepted. However, I still recommend to submit – when possible – the LC-MS/MS raw data to PRIDE repository.
Best regards and good luck.